# Evolving Disaster Response Practices during COVID-19 Pandemic

**DOI:** 10.3390/ijerph18063137

**Published:** 2021-03-18

**Authors:** Gerald Potutan, Masaru Arakida

**Affiliations:** Asian Disaster Reduction Center, Kobe 651-0073, Japan; ma-arakida@adrc.asia

**Keywords:** disaster response system, COVID-19, digitalization, dispersed evacuation, psychological first aid, multihazard approach

## Abstract

When a disaster occurs during a pandemic, it would be a case of concurrent crises (synonymous to cascading disasters or compounding disasters). The single-hazard approach to disaster response system is not suited for this scenario. As shown from the experiences of the Philippines, India, Japan, and the Republic of Korea, the conventional single-hazard approach needed to integrate new measures, including basic skills training on handling COVID-19 for disaster responders; additional stockpiles of face coverings, disinfectants, tents, and personal protective equipment (PPE); social distancing at evacuation centers; updating of standard operation procedures (SOPs) and guidelines for disaster response to adapt to the concurrent crises situations. Building on the reports presented by the member countries of Asian Disaster Reduction Center (ADRC), this paper highlights three evolving disaster response practices during the COVID-19 pandemic: (i) digitalization of some aspects of disaster response, including early warning, surveillance, and impact assessment; (ii) dispersed evacuation to enforce social distancing, including other measures such as testing, tracing, and isolating infected individuals; (iii) remote psychological first aid to disaster-impacted individuals who are already experiencing anxieties from the pandemic. Indicative outcomes of the evolving response practices are discussed, including whether these could serve as entry points to transition the disaster response system from a single-hazard approach towards a multihazard approach.

## 1. Introduction

Multiple disaster events could occur simultaneously in one location. If the disaster response system is ill-prepared, it could result in wide-ranging impacts and greater loss of lives and properties. One example of this is the Great East Japan Earthquake of 2011, where a magnitude 9.0 earthquake caused high tsunami waves of up to 38 m high that subsequently triggered a nuclear power plant failure [1]. Some experts describe such event as cascading disasters, which progresses from one disaster event to a range of interconnected events causing wider impacts with devastating consequences [2]. Since the triple-disaster occurred simultaneously in the same region, the response system was overwhelmed in such a way that designated evacuation centers were not enough to accommodate victims needing shelter, as the number of impacted people was tremendous; stockpiles of foods, blankets, flashlights, and heaters were insufficient; the number of local responders to be dispatched for search and rescue became inadequate [3]. Another example is Typhoon Haiyan that impacted the Philippines in 2013, where a powerful typhoon (with maximum sustained winds of up to 315 kph with gusts of up to 379 kph) triggered a storm surge (of up to 4 m high) and flooding [4]. In this cascading disaster event, the storm surge was believed to be responsible for the greatest number of casualties [5]. The surge swept away people who were pre-evacuated to designated school buildings as well as stockpiles of foods and other emergency goods, as these were situated near the coastal areas [4,5]. Therefore, in the immediate aftermath of the typhoon, the disaster response was overwhelmed with many impacted people remaining unattended for days due to the devastation of communication, roads, and transport systems [4].

The probable occurrence of cascading disasters is higher in Asia, as this region remains the most disaster-prone in the world [6,7]. In view of this, the interest for deeper understanding of cascading disasters has grown, including in research, and learning events [2]. Moreover, the Asian Development Bank (ADB) has integrated a window that helps address cascading disasters (referring to it as *compounding disasters*) in their financing and insurance-related instruments [8]. Furthermore, the Sendai Framework for Disaster Risk Reduction 2015–2030 recognizes compounding disasters from natural hazards and man-made accidents (e.g., industrial accidents) as well as from environmental, technological, and biological hazards (e.g., infectious diseases and pandemics), and therefore, encourages governments to adopt multihazards and multisector approaches in their respective disaster response systems [9]. However, not many Asian countries have adopted a multihazard approach [10], as evident in most disaster response systems that remain to focus on a single-hazard approach [11,12].

### 1.1. Concurrent Crises

In the context of the COVID-19 pandemic, the issue of cascading/compounding disasters is again highlighted in learning events to further inform disaster response policies and actions [13,14]. Since the COVID-19 pandemic is dynamic (considering the emergence of different variants, such as the UK, New York, and Brazil variants) and prolonged-crisis, it could certainly intersect with disasters caused by natural hazards (e.g., earthquake and typhoon). When this happens, it will be a case of cascading/compounding disaster. Yet, recently, some experts also refer to this as concurrent crises, where a disaster that is triggered by natural hazard overlaps with a pandemic that is triggered by a biological hazard [15]. Although the concept of concurrent crises shares similar meaning with both cascading disaster and compounding disaster, this paper adopts the term concurrent crises to put emphasis on the existence of different response systems (i.e., response system for a disaster caused by natural hazard and response system for a crisis caused by biological hazard (e.g., COVID-19)).

### 1.2. Objectives

This paper aims to showcase some of the evolving disaster response practices in Asian countries during the COVID-19 pandemic. It intends to describe how these practices evolve, what are its impacts, and discuss whether these could be entry-points to further improve the disaster response system suited for concurrent crises. In describing the evolving practices, adaptation and changes in the conventional disaster response practices will be highlighted. In describing the impacts, the outcomes of the evolving response practices in preventing the spread of coronavirus infection in disaster-impacted areas will be highlighted. Finally, in describing the implications of these evolving practices for the improvement of the disaster response system, the opportunity to transition from a single-hazard approach towards a multihazard approach will be highlighted.

### 1.3. Materials

To achieve the objectives, the authors utilized the presentations and reports shared by the member countries of the Asian Disaster Reduction Center (ADRC) during the Asian Conference on Disaster Reduction (ACDR) in October 2020 (Appendix A) [16]. In 2020, many Asian countries were impacted by large-scale disasters amid the COVID-19 pandemic. Powerful typhoons, including *Vongfong* (Ambo) in May, *Hagupit* (Dindo) in July, *Molave* (Quinta) and *Goni* (Rolly) in October, and *Vamco* (Ulysses) in November impacted the Philippines. Cyclone *Amphan* in May impacted India. Record-breaking heat waves as well as torrential rains during the summer were followed by huge typhoons, including *Maysak* and *Haishen* both in the month of September, impacting Japan and the Republic of Korea. To know how the disaster response systems, adapt to these concurrent crises, the ACDR organized a session that dealt with disaster preparedness and response measures amidst the COVID-19 pandemic, where key government officials were invited to share their respective experiences. 

## 2. Challenges in Disaster Response during a Pandemic

Recent experiences indicate that the disaster risk management (DRM) agencies in many Asian countries are expected to face unprecedented challenges in dealing with concurrent crises or a disaster on top of the COVID-19 pandemic.

### 2.1. Coordinating Parallel Response Systems

In most countries, the public health emergency response system is separate from the disaster response system. The health agency leads the COVID-19 response while the DRM agency leads the disaster response. These parallel response systems create administrative bottlenecks, including issues concerning leadership, mandates, and coordination. For instance, DRM agency is likely to experience coordination issues in screening and isolating COVID-19 infected individuals at the evacuation centers, if testing and information about quarantined and suspected individuals are under the mandate of the health agency. Having separate response systems for health emergency and for disaster emergency is a common practice in governments that distinguish between biological hazards and natural hazards [10]. Distinguishing biological hazards from natural hazards influences the type of intragovernmental coordination. In a study among 15 Asia-Pacific countries [17], it was observed that there are three types of intragovernmental coordination in responding to COVID-19: (i) the state directly heading an interagency task force, i.e., leadership by a President or Prime Minister, or another direct line of leadership; (ii) the Ministry of Health leading or coordinating the response; (iii) the DRM agency leading or coordinating the response (Table 1). 

Among the 15 countries, five have shown a unified COVID-19 and disaster response system, which is headed by the DRM agency. In this unified response arrangement, the DRM agency coordinates all response measures for concurrent crises, and believed to minimize bottlenecks of intragovernmental coordination. By contrast, having two divergent response systems (one for pandemic and another for disaster) would likely operate in silos, and will be needing an additional mechanism to make the two systems mutually reinforcing. 

### 2.2. Readiness of Disaster Response System

Evidently, a single-hazard response system would be inadequate to effectively respond to concurrent crises. Such system is not ready for concurrent crises as demonstrated earlier in the case of the Great East Japan Earthquake and Typhoon Haiyan. Therefore, in the context of a pandemic, the single-hazard disaster response systems need to put in place additional measures or mechanisms so that they could continue to operate even in the midst of movement restrictions (e.g., border closures and stay-at-home order), social distancing, and face-mask mandates to prevent the spread of infection in disaster-impacted areas. When these limitations are revealed during the actual response activities, the DRM agency needs to immediately adapt its response system to address the challenges. This means that the DRM agency must update its standard operation procedures, protocols, and guidelines for disaster response (Figure 1). 

Adaptation of the response system entails that early warning may now integrate information about clusters/hotspots, outbreaks, and location of COVID-19 infected individuals; emergency stockpiles may now include face masks, disinfectants, testing kits, thermometers, personal protective equipment (PPEs), and vaccines; disaster evacuation procedures may now include testing/screening evacuees, isolation facilities for infected evacuees, and social distancing measures; disaster responders may now require basic knowledge about COVID-19 as well as wear PPEs to prevent the spread of infection in disaster-impacted areas. 

## 3. Evolving Practices

Experiences shared by Asian countries at the ACDR highlighted three evolving disaster response practices, which resulted from the adaptation to the concurrent crises.

### 3.1. Digitalization of Early Warning

Many DRM agencies have accelerated the utilization of digital technologies for disaster early warning, surveillance, and impact assessment to adapt to movement restrictions imposed during the pandemic. The practice of using a digital disaster surveillance dashboard is being enhanced to include information about tracing coronavirus infected individuals as well as contact tracing. Together with the existing data on disasters, COVID-19 information is geo-tagged with local outbreaks in a way that local officials and stakeholders can visualize human behavior through the dashboards. 

In India, the National Disaster Management Authority (NDMA) reported initiating an online dashboard, *National Migrant Information System (NMIS)*, which is hosted on the NDMA-GIS portal [18]. The NMIS maintains a national database, which is useful in facilitating a smooth interstate movement of migrant workers. In maintaining the NMIS, every state collects standard data of each migrating person (i.e., name, age, mobile number, originating district, destination district, and date of travel) to be uploaded on the portal. States can upload batches of files of individual data that are integrated through the Application Programming Interface (API). Additionally, the system generates a unique ID for each migrant to be used for all transactions. In times of disasters, the State Disaster Management Agencies (SDMAs) and other relevant stakeholders can visualize the movement of migrants, and this is useful in informing their respective response decisions. The NMIS is also useful in COVID-19 response by utilizing the mobile data, along with the unique IDs of migrating people, in contact tracing and movement monitoring. By using NMIS information, the dashboard offers real-time analysis report of COVID-19 patient’s case history, which is geo-tagged with information about treatment facilities, food, and shelter homes. Additionally, the dashboard is integrated with various analytics tabs like *Risk Index, Doubling Rate,* and *Quarantine Alerts* [18].

A number of private organizations support the DRM agencies in enhancing their digital portals. The University of Tokyo, for instance, reported an open-source software called *Mobipack* (Spatial Data Commons. Tokyo, Japan) that can be used to monitor population movement. By using the telecommunications’ big data, *Mobipack* accesses the Call Detail Record (CDR) of all mobile phone subscribers to track and map out their movement behavior. Like the NMIS, this shows near real-time movement of people that can be visualized in dashboards and screen monitors. It is useful in responding to disasters during a pandemic, as it can show the hotspots, clusters, and whether people are following the movement restrictions imposed by the government [19]. Another example of digital support is reported by GeoThings (an information software/application and data platform) (GeoThings Inc. Chubei, Hsinchu, Taiwan) that highlighted the enhancement of its disaster response apps called *geoBingAn* to support COVID-19 response. This apps uses Google Forms with OpenStreetMap features on mobile phones to show a *Mask Map* [20]. Through the *geoBingAn*, the mobile user can visualize raw data of whether face masks are available in pharmacies or designated facilities. The *Mask Map* guides people as well as facilitates information exchange about the pandemic and disaster. Specifically, it allows users to find basic information about nearby distributors and their current supply levels as well as other essential data (e.g., contact information and dates when the masks will available). 

In addition to digitalization of early warning, surveillance, and assessment efforts, training activities on disaster response are also increasingly done online. The Singapore Civil Defence Force (SCDF) reported using virtual reality in disaster response training [21]. Likewise, the Ministry of the Interior and Safety (MOIS) of the Republic of Korea reported to have accelerated its online disaster response training, and lessened the conduct of face-to-face training [22].

### 3.2. Dispersed Evacuation

To adapt to social distancing measures, the conventional evacuation practice evolves towards *dispersed evacuation*, enforcing a 2-m distance between individuals [23]. Social distancing at the evacuation centers is a tedious job, as conventional evacuation procedures will no longer be sufficient. Based on the “sphere criteria” (international standard of living environment at the evacuation centers), each evacuee needs a minimum area of 3.5 m^2^ [24]. Now, if social distancing is enforced, each evacuee will require about 6 m^2^ of space at the evacuation centers during a pandemic. This reduces the capacity of any existing evacuation center to almost 50%. In March 2020, Japan reported that due to torrential rain, local residents in Shibecha, Hokkaido evacuated to the gym. Although the gym was originally designed to accommodate 500 evacuees, it could only accommodate 200 evacuees due to social distancing [25]. Likewise, during the devastating torrential rain in the summer of 2020, the public gym in Yatsushiro, Kumamoto Prefecture could only accommodate about half of its original capacity due to social distancing (Figure 2). 

The implications of dispersed evacuation include the need to designate more evacuation centers, identify isolation facilities for infected individuals, and implement additional health measures, including stockpiling of additional emergency supplies (e.g., facemask, disinfectants, and thermometers). 

#### 3.2.1. More Evacuation Centers

Usually, DRM agencies designate schools as evacuation centers. However, during the pandemic, those designated schools would no longer be enough. In the Philippines, churches and stadiums were used as evacuation centers during typhoon *Vongfong* [16]. In India, some evacuees from cyclone *Amphan* were accommodated in marriage halls and other public buildings [16]. In Japan, individuals who needed to evacuate during the torrential rains were also encouraged to stay in their cars (but away from vulnerable areas) or go to their relative’s house [26]. 

#### 3.2.2. Isolation Facilities

Securing an isolation facility is essential to further prevent the spread of infection. In the Philippines, the Office of Civil Defense (OCD) reported designating isolation facilities for COVID-19 infected evacuees, which the government considered as good practice in responding to the impact of typhoon *Vongfong* in May 2020 [16]. Administratively, there are at least two common procedures of identifying who shall stay in the isolation facility. One procedure is by acquiring the records from public health office about the following individuals: under quarantine, traced to have close contact with infected individuals, returnees from abroad, and known to have visited cluster or outbreak areas. Another procedure is by testing and screening evacuees before arrival at the evacuation center. In performing these procedures, it is necessary for the DRM agency to coordinate with the public health agency to ensure mutual reinforcement of functions. 

#### 3.2.3. Additional Measures 

While at the evacuation centers, disaster responders need to check the health conditions of the evacuees as well as promote washing of hands, use of facemasks, and practicing coughing etiquette. It is also important to maintain a hygienic environment as well as ensure adequate ventilation. Furthermore, it is recommended to secure a private space for people who show symptoms (e.g., cough and fever) as well as ensure that they could use separate toilets. As shown in earlier experiences, DRM agencies were not ready for these additional measures [27]. The lack of preparation was evident when disaster responders used cardboard, instead of tents, as partitions at the evacuation centers. In order to effectively practice dispersed evacuation in the future, DRM agencies need to prepare in advance, especially in stockpiling of needed supplies like facemasks, tents, thermometers, disinfectants, PPEs, and partition materials. With the inevitable intersection between DRM responders and health responders during concurrent crises, preparedness efforts in these two fields may be coordinated pre-disaster to effectively respond to an eventual mass casualty incident. One insight from the medical field is that pre-training gap analysis and identification of competencies and skills of responders should be conducted prior to training them in mass casualty incidents and disasters, as this offers opportunity to develop training curriculum at various skill and knowledge levels [28].

### 3.3. Remote Psychological First Aid

Apart from the worries associated with lockdowns, movement restrictions, quarantines, physical distancing, and closures of schools and establishments, the COVID-19 pandemic has increasingly triggered more anxieties, such as losing loved ones, becoming infected, and to be denied at the overloaded hospitals [29]. These worries are further exacerbated during disasters by having anxieties of living in evacuation shelters or losing personal properties. Therefore, offering psychosocial support to disaster-impacted individuals during the pandemic has gained traction to make people calm during concurrent crises. In India, the NMDA reported that it enhanced its psychosocial support measures by providing counseling to COVID-19 patients [16]. In this initiative, NDMA developed a manual, Psychosocial Support for People Testing Positive for COVID-19 to serve as guide in achieving two main objectives: ‘lend a listening ear’ and ‘speak comforting word’ to inspire hope and positivity. In implementing this initiative, NDMA partnered with a private company (Tata Institute of Social Sciences) in setting up a psychosocial hotline and mobilization of volunteers. Due to movement restrictions, the support, usually in a form of psychological first aid (PFA), is offered remotely. The psychosocial hotline can be contacted through a mobile app that both the volunteers and patients could download. The first step to becoming a volunteer is to enroll for the program at NDMA. Once accepted, the volunteer will be given an orientation on the process of counseling. Certified volunteers will then install the mobile apps for counselling services and dedicate at least two hours daily or select days of the week to work. NDMA emphasized that volunteers are not expected to make calls to reach out for patients, but the other way around. As volunteer counsellors need to know the name, age, gender, district, state, and phone numbers of the patients, they must strictly ensure privacy and confidentiality. Each counselor must also send a brief report of each conversation with the patient using a format specified by the NDMA. The Republic of Korea has likewise reported to have enhanced its psychosocial support measures in responding to disasters during the pandemic [16]. 

## 4. Discussion

Outcomes of the evolving practices indicated not only to have prevented the spread of the coronavirus infection, but it also reduced the loss of lives and damage to properties in disaster-impacted areas. The DRM agencies of the Philippines, India, Japan, and the Republic of Korea reported that by enforcing the evolving disaster response practices, they were able to prevent the spread of the coronavirus infection and avoided a spike of cases [16]. More importantly, the outcomes appear to validate the importance of multisector and multihazard approaches in responding to concurrent crises. While disaster response practices in digitalization, dispersed evacuation, and remote psychological first aid are still evolving, the actual cases presented in this paper suggest that the evolution of these practices is gearing towards a more robust disaster response system. 

### 4.1. Mobilizing Multisector Approach

Responding to concurrent crises involves a range of government agencies with different systems such that the health agency leads in tracking infections, tracing contacts, or determining clusters/hotspots while the DRM agency leads the evacuation, search and rescue, and relief efforts. As shown in the cases, these systems often operate in silos, and it causes bottlenecks whenever the two systems intersect. For instance, an effective practice of dispersed evacuation requires the integration of data from the health agency and the DRM agency to reduce coordination bottlenecks. The implication of this observation is that it validates the importance of mobilizing a multisector approach to respond to concurrent crises. At the ACDR [16], the SCDF reported that the national government adopted the Whole-of-Government (WOG) approach in responding to the COVID-19 pandemic. In this approach, all agencies of the government jointly respond to the pandemic, including disasters that may occur, to work for a common solution. In view of this approach, the SCDF continues to operate in “business-as-usual” (i.e., responding to every day emergency calls and in standing ready to render humanitarian assistance to disasters in the region) while stepping up support to the national COVID-19 response. Moreover, in Nepal, where federalism is recently ratified as the form of government, the National Disaster Risk Reduction and Management Authority (NDRRMA) is coordinating with the Ministry of Health and Population (MoHP) to harmonize pandemic response measures with disaster response measures in the three tiers of the government (i.e., at the federal, provincial, and local levels). 

### 4.2. Adopting a Multihazard Approach

Evidently, as shown in the experiences of India, Japan, and the Philippines, biological hazards are not yet fully integrated in the DRM systems. Therefore, preparedness measures and information needed to respond to a disaster during a pandemic are not yet fully put in place. Moreover, the focus on the single-hazard approach prior to the pandemic hindered the DRM systems to integrate response measures that are adapted to movement restrictions and social distancing. As observed from the cases, the supplies of tents, facemasks, disinfectants, isolation facilities, and partitions were not enough at the evacuation centers because the single-hazard approach of disaster response system was ill-prepared for multiple events. Obviously, the multihazard approach could have been better for concurrent disasters. 

In recognizing the limitation of the single-hazard approach, some DRM agencies of ADRC member countries are now reviewing their respective disaster response plans and programs with the consideration of adopting a multihazard approach [16]. In Armenia, the Ministry of Emergency Situations (MES) reported that it is reviewing their risk assessment methodologies as well as its procedures for risk analysis to integrate all other risks, including biological hazards. After the review, the MES will subsequently update its Emergency Response Plan to adapt to multirisk environments. In Tajikistan, the Committee of the Emergency Situations (CoES) reported piloting *Risk Changes*, an open-source tool for multihazard risk assessment, with the University of Twente’s Geo-Information Science and Earth Observation (ITC). This tool allows the end-users and stakeholders to assess and evaluate the prevailing risks in a designated area and decide the best available risk reduction alternatives. It analyzes not only the changes in hazards but also the changes taking place in elements-at-risk (e.g., buildings, population, and transportation infrastructure), especially its vulnerabilities. 

While not reported in the ACDR, it is important to note that in 2019, the World Health Organization (WHO) put together a Health Emergency and Disaster Risk Management Framework (H-EDRM), which is a useful document for adopting a multihazard approach. The H-EDRM emphasizes preparing for all hazards, focusing on societies’ vulnerabilities and capabilities, mobilizing the whole society and not just the health sector in responding to disasters [30]. 

### 4.3. Using Lessons on Evolving Practices to Inform Policies and Programs

Most of the evolving disaster response practices during a pandemic could inform the improvement of DRM systems to effectively respond to concurrent crises. At the national level, the disaster management acts of respective countries in Asia may be amended to cover all hazards (natural, man-made, and biological), and mandate the DRM agency to coordinate all response measures, including pandemic. In view of this, it would be useful to learn from the experiences of Indonesia, New Zealand, Samoa, Tonga, and Vanuatu since their respective DRM agencies coordinate both COVID-19 response and disaster response [17]. In improving programs at the local level, it is useful to learn from the lesson of engaging the community such as serving as volunteers in rendering remote psychological first aid during concurrent crises. PFA is a well-established approach of helping disaster-affected individuals. Therefore, like in India, DRM agencies can strengthen PFA by doing it remotely with the support of volunteers from community. In essence, PFA is about facilitating stabilization, grounding, and a return to daily functioning in the immediate aftermath of a crisis [28]. Therefore, it is not a therapy, but a simple intervention by lending a listening ear, assurance, encouragement, and acknowledgement of experiences in order to normalize reactions to the abnormal events. Learning from the experience of NDMA-India, remote PFA can be accelerated through mobile apps, social media platforms, or telephone/online calls. One advantage of remote PFA is that personal information could be easily disclosed. 

## 5. Conclusions

All cases of evolving disaster response practices during a pandemic, as presented in this paper, point to the following conclusions. Firstly, during concurrent crises, the DRM agency is not independently leading the response efforts. Instead, there are two separate response systems that operate: disaster response system and the COVID-19 response system. As observed in the practice of dispersed evacuation, the health system and the DRM system often operate in silos, and this creates logistical bottlenecks. To lessen this, another mechanism is needed to make the systems mutually reinforcing. One of the mechanisms that might be insightful in improving the intragovernmental coordination is the Whole-of-Government (WOG) approach. In this regard, it could be said that the evolving disaster practices presented in this paper are evidence that a multisector approach is important in responding to concurrent crises.

Secondly, disaster response practices are evolving since the single-hazard approach is ill-prepared for concurrent crises. Hence, it needs to adapt by integrating additional measures (e.g., social distancing and movement restriction) to prevent the spread of infection during a pandemic. These evolving practices of digitalization, dispersed evacuation, and remote PFA validate the need for DRM systems to transition from a single-hazard approach towards a multihazard approach in order to effectively respond to concurrent crises. 

Finally, the lessons from these evolving practices are useful to inform the improvement of DRM policies and programs. The evolving practices validate the need to amend the disaster management acts of respective countries in Asia to cover all hazards (natural, man-made, and biological), and mandate the DRM agency to coordinate all response measures, including a pandemic. The evolving practices also offer insights to local governments to mobilize communities to actively engage in responding to concurrent crises either by volunteering in community-based programs (e.g., remote PFA) and actively supporting the implementation of disaster response measures (e.g., signing up to evacuate to relatives houses to prevent overcrowding at the evacuation centers). Overall, this paper offers actual experiences, and evidence-based cases of evolving disaster response practices.

## Figures and Tables

**Figure 1 ijerph-18-03137-f001:**
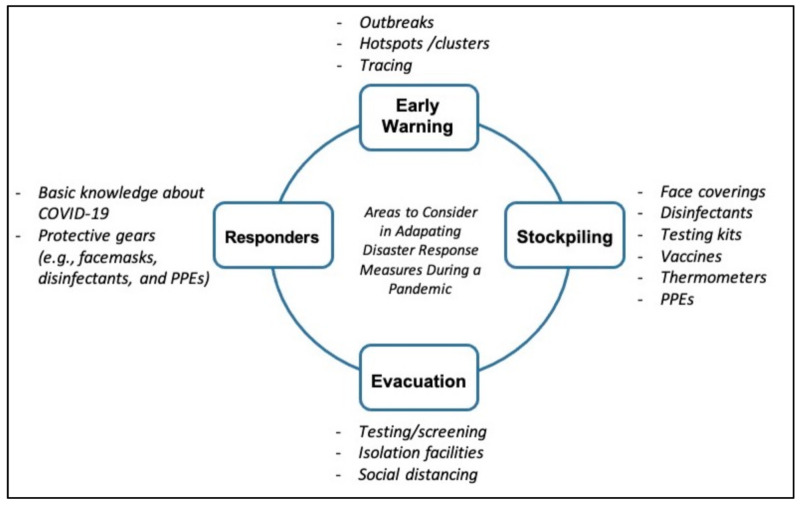
Adapting disaster response system with COVID-19 response.

**Figure 2 ijerph-18-03137-f002:**
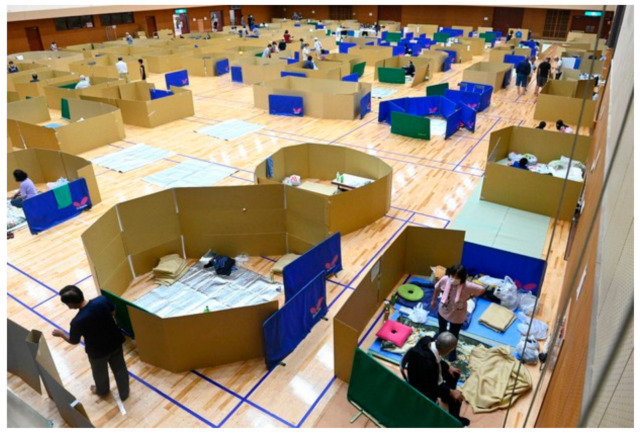
Public gym in Yatsushiro, Kumamoto Prefecture (Mainichi, 2020).

**Table 1 ijerph-18-03137-t001:** Intragovernmental arrangements to COVID-19 response.

Lead	Countries
Task Force (Presided by a President or Prime Minister)	Japan
	Myanmar
	Republic of Korea
	Vietnam
Health Agency (e.g., Ministry of Health)	Cambodia
	Fiji
	Papua New Guinea
	Philippines
	Singapore
DRM Agency (e.g., National Disaster Management Authority)	Indonesia
	New Zealand
	Samoa
	Tonga
	Vanuatu

## Data Availability

This study did not report any data.

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
