# Peer review of "Evolving Disaster Response Practices during COVID-19 Pandemic"

_ijerph, 2021, doi:10.3390/ijerph18063137_

Round 1

Reviewer 1 Report

This paper present a review about "Evolving Disaster Response Practices During COVID-19 Pandemic". This paper is well written the the subject is very actual. I only suggest that in the beginning of the introduction a more complete state of the art of this subject is presented and at the end the objective of this paper is described in detailed. 

Indeed, what I suggest is that an "introductory section is created describing the current state of the art and the inclusion at the end of this introductory section the objectives of this paper".

Author Response

Point 1: State of the Art of the subject

Authors added an introduction section detailing the state of the art.

Point 2: Objective of the paper

Objectives of the paper are elaborated in the introduction section.

Reviewer 2 Report

The article discusses the basic challenges of the current covid-19 pandemic in multi-hazard scenarios. The authors argue about the challenges of the current pandemic in the context of other natural disasters. The article mainly focuses on possible challenges and the suitability of different national models for managing multiple crisis situations.

The article is purely descriptive. In conclusion, the authors suggest possible improvements for multi-hazard situations. However, the authors do not offer any strong arguments to confirm their conclusions.

The presented article cannot be considered a full-fledged research article.

Author Response

Point 1: Offer strong arguments to confirm conclusions

The discussion section of the paper is revised to elaborate the key arguments, and were confirmed in the conclusions.

Reviewer 3 Report

The work is interesting and acceptable for publication considering the relevance at a national scale. However, it needs an improvement.

The Authors should consider also cascading effects. The introduction needs to be enriched with more references also related to cascading effects.

The Authors should clarify in the Introduction whether this is related to trends in Asia or in general. You mention it domestic, so I guess they refer to some countries? but it is not introduced anywhere earlier. Please make it clear since the introduction.

- The title is about the evolving, but the method on how to analyse the evolving is not elaborated clearly. Also, the result of "evolving" is not clearly presented. I suggest to provide temporal basis of the research topic or keywords in this field, so we can see which topic is increasing at a certain time.

The references in the study are limited for a review study. It would be more appropriate to be categorized as a or the references list need to be extensively enriched. Include following reference related to distaster and education
https://doi.org/10.1186/s12889-021-10165-5

  • The data has been presented greatly but is not elaborated adequately.
  • The authors should Focus on discussion the findings since there is no enough discussion of the findings

The presented figures (photos) are hardly legible. I suggest reducing their quantity to a minimum

Author Response

Point 1: Consider cascading effects and clarify the scope of the study

An introduction section is added, detailing the state of the art of cascading disasters and its effects. It also specified the Asian coverage and temporal basis of the study (i.e., disaster during a pandemic).

Point 2: Focus on discussions and findings, and elaborate analysis of how response practices evolve as well as its results

The discussion section is revised to elaborate how disaster response practices evolved, what are its results, and the implications to DRM system.

Point 3. Enrich the references

In improving the discussions, it subsequently enriched the references, including the one suggested by the reviewer.

Point 4: Reduced figures

Number of figures is reduced to 2.

Round 2

Reviewer 1 Report

The authors have satisfactorily answered the referee's suggestions. 

Reviewer 2 Report

The presented paper is interesting and the edits improve its quality. The main issues still stands, since I have a hard time classify this paper as a full-fledged research article. 

Reviewer 3 Report

The author have addressed all concerns raised before.